# Casual Sex and Sexlessness in Japan: A Cross-Sectional Study

**Shoko Konishi** [1],*, **Yoshie Moriki** [2], **Fumiko Kariya** [1] **and Manabu Akagawa** [3]

1   Department of Human Ecology, School of International Health, Graduate School of Medicine,
    The University of Tokyo, Tokyo 113-0033, Japan; kariya@humeco.m.u-tokyo.ac.jp
2   College of Liberal Arts, International Christian University, Tokyo 181-8585, Japan; moriki@icu.ac.jp
3   Graduate School of Humanities and Sociology, The University of Tokyo, Tokyo 113-0033, Japan;
    akagawa@l.u-tokyo.ac.jp
*   Correspondence: moe@humeco.m.u-tokyo.ac.jp; Tel.: +81-3-5841-3487; Fax: +81-3-5841-3395

**Abstract:** Low fertility has persisted in Japan for decades. Sexless marriages may indirectly contribute to low fertility. Inactive sexual lives within intimate and committed relationships may be linked to sexual activity outside such relationships, called "casual sex". This study aimed to explore the correlates of casual sex and sexlessness. A web-based questionnaire survey was conducted among married and single men (*n* = 4000) aged 20–54 years in Japan. Sexlessness were reported by 56% of men, whereas 11% had had casual sex and 31% had had non-casual sex (with spouse, fiancé, or girlfriends/boyfriends) in the last month. Among married men, higher income and long working hours were positively associated with casual sex. Regarding never-married men: those with lower educational status and without full-time jobs were more likely to report casual sex, those in rural areas were more likely to be sexless than those in urban and suburban areas, and those with depression were more likely to be sexless than those without depression. Matching app use was strongly associated with casual sex among married and never-married men, suggesting that such tools may facilitate sexual activity outside committed and intimate relationships. Sexual behavior is closely linked to one's social and economic environment and health status.

**Keywords:** Asia; geographic factors; health; Japan; male; marriage; sexual behavior; social networking; socioeconomic status

## 1. Introduction

Sexless marriages [1] and low fertility [2] are prevalent in Japan. Considering that coital frequency is one of the proximate determinants of fertility [3] (p. 71), sexlessness and low fertility might be linked. The monthly probability of conception was found to be positively associated with coital frequency in Japanese couples trying to conceive their first child [4]. Thus, low coital frequency may be associated with a higher risk of infertility, that is, the inability to become pregnant within one year of unprotected intercourse [5]. A national survey found that of married couples trying to conceive a child, 40% were sexless, that is, they had had no sexual intercourse in the past month [1]. Similarly, among married women who wished to conceive at the time of the survey, 29% were sexless [6]. Despite the importance of the frequency of sexual intercourse as a potential predictor of infertility and fertility, there are few scientific surveys of sexual behavior and its correlates in Japan.

Two national surveys have reported the frequency and correlates of sex among men and women in Japan. In the 2007 National Survey of Work and Family, the correlates of sexless marriages were examined [1]. Factors related to sexless marriages in men included older age, poor health, full-time employment, working 60+ hours per week, having children aged three or younger, stress with spouse, and perceived risk of divorce [1]. Another national survey, the 2000 General Social Survey (JGSS), found that the frequency of sex among single men and women was lower among those who did not have jobs than among those who worked [7]. The JGSS survey also showed that in married employees of both

sexes, the weekly working hours were negatively associated with sex frequency [7]. In addition to these two national surveys, the Japan Family Planning Association conducted a series of surveys targeting Japanese men and women regarding their awareness and behavior related to sexuality [8]. In their 2014 survey, participants were asked about their reasons for not perceiving sex positively. The answers included not having a partner, being bothered, and being tired from work [8] (p. 110).

In the present study, casual sex is defined as sex with a person with whom one is not in an intimate and committed relationship. Non-casual sex partners included spouses (for married men) and fiancés or girlfriends/boyfriends (for single men). The 2014 survey by the Japan Family Planning Association found that 3.2% of male respondents and 2.1% of female respondents had one or more casual sex partner at the time of the survey. According to the Jex Japan Sex Survey [9], conducted online in 2020, 67.9% of men and 46.3% of women reported having engaged in casual sex at some point. The latter survey, but not the former, suggests that casual sex may be relatively common, which contrasts with the prevalence of sexless marriages in Japan [1,6].

To date, no quantitative study in Japan has examined sexual activities by different types of relationships, namely casual or non-casual. In the National Survey of Work and Family, respondents were asked to indicate the frequency of sexual intercourse with their spouse in the past year [1]. In the JGSS, respondents were asked to indicate the frequency of sex (not limited to sexual intercourse) in the past year, without specifying who they had sex with [7]. An important research question of the present study was "In what relationships do Japanese people have sex?" Based on personal communications and other sources, the authors assumed that there are at least seven different types of relationships: spouse/fiancé, boyfriend/girlfriend (*koibito*), more than a friend but less than a girlfriend/boyfriend (*tomodachi ijou koibito miman no hito*), friends with benefits (no committed relationship but having sex with a person with whom one meets regularly; *sex friend*), friend or casual acquaintance (*yujin/chijin*), one-night stand (*ikizuri no hito*), and sex workers or other forms of paid sex (*seifuzoku no hito*). Thus, in the present study, information was obtained on the frequency of sex, indicating the type of sexual partner. As casual sex is more common among men than women in Japan [9] and in other countries [10,11], this study targeted only men aged 20–54 years.

Given that sexless marriage and low fertility may be interrelated, examining what factors contribute to sexlessness and casual sex is important. Sexless relationships may already be present among unmarried couples. We speculate that sexlessness within intimate and committed relationships may be partly compensated for by casual sex. Thus, we explored which social, economic, and health-related factors are associated with sexlessness and casual sex among married and never-married men. We assumed that sex within intimate and committed relationships is more likely to result in pregnancy and marriage, which would lead to reproduction. By contrast, we assumed that casual sex is less likely to result in reproduction. By examining factors associated with casual sex and sexlessness, we aimed to understand who among the Japanese people are more likely to engage in sexual activity that is less likely to result in reproduction.

Casual sex may be facilitated by social networking services (SNSs) and matching apps. In 2020, more than 85% of individuals aged 20–59 years in Japan owned smartphones [12]. Because SNSs and matching apps can be installed and used on smartphones, it makes it easier to meet casual sex partners. Therefore, this study analyzed the relationship between SNSs and matching apps and casual sex. The socioeconomic status and health-related factors were included in the analyses, as they were also considered to be associated with sexual behavior. In a previous study using the JGSS data, long working hours were associated with less frequent sex [7]. Additionally, health-related factors have been shown to be associated with sexual behavior and functioning—depression and erectile dysfunction often present together in men [13,14]. Furthermore, fertility treatment among married couples in Japan was associated with less frequent sex [15]. Consequently, depression and fertility treatment were included in the analyses. This study aimed to (1) collect quantitative

data on sexual behavior by relationship type and (2) explore the correlates of casual sex and sexlessness among men in Japan. Identifying the correlates of sexual behavior can help us to understand the social structure that contributes to widespread sexlessness among married couples in this setting. Considering that pregnancy (but not childbirth) often precedes marriage in Japan [16], the sexual behavior of single men is expected to impact the rates of marriage and fertility in the country.

## 2. Materials and Methods

The Interdisciplinary Investigation of Technology, the Environment, and Fertility (IITEF) project (P.I.: Shoko Konishi) aims to understand how technology and the environment interact to shape fertility trends in contemporary society. The IITEF project consists of two subprojects: a study of the associations between technological innovations and variability in human sexual behavior using web-based questionnaire surveys, and a study of the potential effects of endocrine disruptors on male reproductive function based on biomarker measurements. The ultimate goal of the project is to construct a model that incorporates all variables from the two subprojects to assess their combined impacts on human fertility. As part of the first subproject, two questionnaire surveys (retrospective and cross-sectional) were conducted online in September 2020. This retrospective survey aimed to explore secular changes in the sexual behavior of Japanese men over a period of up to 20 years, whereas the cross-sectional survey aimed to obtain information about sexual activity, focusing on the type of sexual partner. The retrospective and cross-sectional surveys each consisted of 4000 completed questionnaires. For the present study, data from the cross-sectional survey were used, the details of which are described below. Selected questions from the cross-sectional survey are listed in Supplementary Material 1.

The authors contracted Rakuten Insight, Inc. to collect responses from 4000 eligible participants. The eligibility criteria required participants to be male, aged 20–54 years, and residing in Japan. The final number (total $n = 4000$) of respondents by 5-year age category and eight geographic regions were set to be relative to the corresponding proportions of the male population in Japan in the 2015 National Census (Supplementary Table S1). Across the country, there were more than two million registered Rakuten members, which included 661,604 eligible men. A weblink to the online questionnaires was sent to eligible members by email. The questionnaires consisted of the informed consent, three screening questions, and 42 main questions. The screening section sought details about participants' age, marital status, and prefecture of residence (Supplementary Material 1). The weblink to the questionnaire was closed when the number of respondents in each subgroup (by age category and geographic region) exceeded the set number. As a result, the questionnaires were sent to 75,118 eligible men, among whom 4858 provided complete sets of responses. Respondents who were suspected of sharing low-quality answers (e.g., the response time was either too short or too long) were excluded using Rakuten Insight's algorithm. The number of excluded respondents is unknown. From the remaining samples ($n < 4858$; exact number unknown), data from a total of 4000 respondents (Supplementary Table S1) were randomly selected and provided to the authors, which were then analyzed.

### 2.1. Sexual Behavior

Information on sexual behavior was collected using the questions listed in Supplementary Material 1. Participants were asked whether they had ever had sex. Those who responded in the affirmative were asked whether they had had sex in the last month. Those who had had sex in the last month were asked about the number of sex partners and were asked to select the type(s) of partner(s) they had sex with. There were eight choices: spouse/fiancé, girlfriend/boyfriend (*koibito*), more than a friend but less than a girlfriend/boyfriend (*tomodachi ijou koibito miman no hito*), friend with benefits (not in a committed relationship but having sex regularly with a person; *sex friend*), friend or casual acquaintance (*yujin/chijin*), one-night stand (*ikizuri no hito*), sex worker or other forms of paid sex (*seifuzoku no hito*), and other.

Participants were also asked to select the frequency of sex in 2019 from among 10 choices. The frequency of sex in 2019 was investigated because the authors wanted to avoid the potential effect of COVID-19, which spread after January 2020 and involved restrictions and changes in people's behavior.

### 2.2. Demographic, Socioeconomic, Health-Related Variables, and Use of SNS and Matching Apps

Data on the age, marital status, educational status, job status, annual income, and weekly work hours were collected (Supplementary Material 1). Participants were also asked to select their self-rated health status. They were asked whether they had ever been diagnosed with depression by a medical doctor, as well as about their habit of smoking cigarettes. Whether they had ever visited a hospital for consultation and/or treatment for their or their partner's infertility issues was also asked. Participants were asked to select SNS services and the matching apps they had used in the past month. The answer choices were TikTok, LINE, Twitter, Facebook, Instagram, and matching apps (Supplementary Material 1).

### 2.3. Place of Residence

Using the seven-digit postal code of the participants' home addresses, names of geographic units smaller than towns or villages were obtained. Each geographic unit was then matched with one or more census enumeration unit (>1.8 million in the country) used in the 2015 census. Each census enumeration unit is categorized as a densely inhabited district (DID), semi-densely inhabited district (semi-DID), or non-densely inhabited district (non-DID) by the Ministry of Internal Affairs and Communications [17,18]. Geographic units whose matching census units consisted only of DIDs were categorized as urban, while those consisting exclusively of non-DIDs were categorized as rural. Other geographic units that consisted of a mixture of DID, semi-DID, or non-DID census units were categorized as suburban. A total of 19 zip codes could not be matched to their corresponding census enumeration units, for example, due to changes in municipal boundaries between the 2015 Census and the time of the survey. In addition, 355 participants refused to respond to the zip codes, and 92 participants responded to zip codes that did not exist. In total, 466 participants had missing values for their place of residence.

### 2.4. Statistical Analyses

Sample sizes by age group and marital status were tabulated for the present participants and compared with data from the 2020 census in Japan [19]. To calculate summary statistics, participants were divided into three groups: married, never married, and other marital status. Because it was expected that correlates of sexless relationships and casual sex would differ by marital status, statistical analyses were conducted separately for married and never-married men. Men with other marital statuses were excluded from further analysis because of the small sample size.

Sexual behavior in the last month was divided into three groups: sexless, non-casual sex, and casual sex. If a participant (either married or single) reported not having had sex in the past month, he was classified as sexless. If a married man reported having had sex with his spouse in the past month, he was categorized as having non-casual sex. If a married man reported having had sex in the last month but not with his spouse, he was categorized as having casual sex. If he had both casual and non-casual sex, he was categorized as having non-casual sex in the statistical analyses. We did not set a category that included having both casual and non-casual sex because of the small sample size. For a single man, casual sex was defined as sex with a partner other than a fiancé or a girlfriend/boyfriend.

Multinomial logistic regression analyses were performed to examine the associations between possible covariates and sexual behavior. The following analyses were conducted separately for married and never-married men. The reference category was defined as having non-casual sex. The relative risk ratios (RRRs) and 95% confidence intervals (CIs) for sexlessness and casual sex were estimated. The first multinomial logistic regression analysis included age and a predictor variable in each model. The second set of analyses included

all covariates in the model simultaneously. Predictor variables for married men included self-rated health (good or other), depression (no or yes), fertility treatments (no or yes), cigarette smoking (current smoker or other), education (less than university, or university or higher), occupational status (full-time or other), log annual income, weekly hours worked (<50, 50–59, or 60+), place of residence (rural, suburban, or urban), and the use of information tools (yes or no for each of the following: TikTok, LINE, Twitter, Facebook, Instagram, and matching apps). Predictor variables for never-married men were the same as those for married men, except that fertility treatment was not included. It is quite rare for never-married men to undergo fertility treatment. These predictor variables were selected because the health status, health-related behaviors, and socioeconomic status are expected to influence motivation, libido, and/or the opportunity to have sex. Generalized variance inflation factors (GVIFs) were used to confirm that there were no multicollinearity problems in the multinomial logistic regression analyses. All statistical analyses were conducted using R ver.4.1.1 [20].

## 3. Results

Of the 4000 respondents, 2035 (51%) were married; 1701 (43%) were never married; and 264 (7%) were divorced, widowed, or of another marital status (Table 1). The proportion of married men (51%) was slightly higher than that in the 2020 census (49%). The distribution of age categories was the same in the present sample and in the 2020 census (Table 1). The present sample's mean (SD) age was 38.5 (9.9) years. Among married men, 15% had visited a hospital for consultation and/or treatment of fertility issues. The use of dating apps in the past month was reported by 2% of married men and 9% of never-married men (Table 2).

**Table 1.** Sample sizes by age group and marital status of the present sample and of the 2020 national census of Japan (male population only).

| Age Group | Total | Single | Married | Other | Proportion of Age Group | Single | Married | Other |
|---|---|---|---|---|---|---|---|---|
| The present sample | | | | | | | | |
| 20–24 | 481 | 439 | 28 | 14 | 12% | 91% | 6% | 3% |
| 25–29 | 470 | 318 | 135 | 17 | 12% | 68% | 29% | 4% |
| 30–34 | 503 | 216 | 257 | 30 | 13% | 43% | 51% | 6% |
| 35–39 | 558 | 179 | 357 | 22 | 14% | 32% | 64% | 4% |
| 40–44 | 641 | 226 | 375 | 40 | 16% | 35% | 59% | 6% |
| 45–49 | 720 | 197 | 450 | 73 | 18% | 27% | 63% | 10% |
| 50–54 | 627 | 126 | 433 | 68 | 16% | 20% | 69% | 11% |
| Total | 4000 | 1701 | 2035 | 264 | 100% | 43% | 51% | 7% |
| 2020 national census [1] | | | | | | | | |
| 20–24 | 3,233,994 | 3,095,077 | 131,245 | 7672 | 12% | 96% | 4% | 0% |
| 25–29 | 3,279,149 | 2,505,473 | 744,487 | 29,189 | 12% | 76% | 23% | 1% |
| 30–34 | 3,431,250 | 1,776,898 | 1,583,104 | 71,248 | 13% | 52% | 46% | 2% |
| 35–39 | 3,805,952 | 1,463,512 | 2,213,585 | 128,855 | 14% | 38% | 58% | 3% |
| 40–44 | 4,298,675 | 1,385,680 | 2,713,344 | 199,651 | 16% | 32% | 63% | 5% |
| 45–49 | 4,993,896 | 1,491,312 | 3,195,408 | 307,176 | 18% | 30% | 64% | 6% |
| 50–54 | 4,394,401 | 1,170,416 | 2,878,505 | 345,480 | 16% | 27% | 66% | 8% |
| Total | 27,437,317 | 12,888,368 | 13,459,678 | 1,089,271 | 100% | 47% | 49% | 4% |

[1] Statistics Bureau of Japan [19], male population only.

**Table 2.** Basic characteristics of participants by marital status (*n* = 4000).

| Variables | Married (*n* = 2.035) | Never Married (*n* = 1.701) | Other (*n* = 264) | Total (*n* = 4000) |
|---|---|---|---|---|
| **Age (years)** | | | | |
| Mean (SD) | 42.0 (8.0) | 33.7 (9.9) | 42.5 (9.1) | 38.5 (9.9) |
| Median [Min, Max] | 43.0 [20.0, 54.0] | 32.0 [20.0, 54.0] | 45.0 [20.0, 54.0] | 39.0 [20.0, 54.0] |
| **Self-rated health** | | | | |
| Good | 1810 (89%) | 1447 (85%) | 227 (86%) | 3484 (87%) |
| Poor | 225 (11%) | 254 (15%) | 37 (14%) | 516 (13%) |
| **Depression** [1] | | | | |
| No | 1902 (93%) | 1562 (92%) | 232 (88%) | 3696 (92%) |
| Yes | 133 (7%) | 139 (8%) | 32 (12%) | 304 (8%) |
| **Fertility treatment** | | | | |
| No | 1720 (85%) | 1684 (99%) | 239 (91%) | 3643 (91%) |
| Yes | 315 (15%) | 17 (1%) | 25 (9%) | 357 (9%) |
| **Smoking** | | | | |
| No | 1509 (74%) | 1375 (81%) | 169 (64%) | 3053 (76%) |
| Yes | 526 (26%) | 326 (19%) | 95 (36%) | 947 (24%) |
| **Educational status** | | | | |
| Less than university | 710 (35%) | 706 (42%) | 142 (54%) | 1558 (39%) |
| University or higher | 1317 (65%) | 986 (58%) | 120 (45%) | 2423 (61%) |
| Missing | 8 (0%) | 9 (1%) | 2 (1%) | 19 (0%) |
| **Job status** | | | | |
| Full-time | 1835 (90%) | 959 (56%) | 165 (63%) | 2959 (74%) |
| Other | 200 (10%) | 742 (44%) | 99 (38%) | 1041 (26%) |
| **Annual income (million yen)** | | | | |
| Mean (SD) | 6.62 (3.85) | 3.44 (3.79) | 5.56 (5.84) | 5.22 (4.27) |
| Median (Min, Max) | 5.50 [0, 50.0] | 3.50 [0, 50.0] | 4.50 [0, 50.0] | 4.50 [0, 50.0] |
| Missing | 115 (6%) | 143 (8%) | 22 (8%) | 280 (7%) |
| **Weekly work hours** | | | | |
| <50 | 1257 (62%) | 1293 (76%) | 170 (64%) | 2720 (68%) |
| 50–59 | 480 (24%) | 214 (13%) | 49 (19%) | 743 (19%) |
| 60+ | 262 (13%) | 109 (6%) | 35 (13%) | 406 (10%) |
| Missing | 36 (2%) | 85 (5%) | 10 (4%) | 131 (3%) |
| **Place of residence** | | | | |
| Rural | 248 (12%) | 198 (12%) | 32 (12%) | 478 (12%) |
| Suburban | 585 (29%) | 428 (25%) | 66 (25%) | 1079 (27%) |
| Urban | 979 (48%) | 862 (51%) | 136 (52%) | 1977 (49%) |
| Missing | 223 (11%) | 213 (13%) | 30 (11%) | 466 (12%) |
| **TikTok** [2] | | | | |
| No | 1907 (94%) | 1524 (90%) | 239 (91%) | 3670 (92%) |
| Yes | 128 (6%) | 177 (10%) | 25 (9%) | 330 (8%) |
| **LINE** [2] | | | | |
| No | 264 (13%) | 386 (23%) | 60 (23%) | 710 (18%) |
| Yes | 1771 (87%) | 1315 (77%) | 204 (77%) | 3290 (82%) |
| **Twitter** [2] | | | | |
| No | 1241 (61%) | 653 (38%) | 166 (63%) | 2060 (52%) |
| Yes | 794 (39%) | 1048 (62%) | 98 (37%) | 1940 (49%) |
| **Facebook** [2] | | | | |
| No | 1246 (61%) | 1230 (72%) | 180 (68%) | 2656 (66%) |
| Yes | 789 (39%) | 471 (28%) | 84 (32%) | 1344 (34%) |
| **Instagram** [2] | | | | |
| No | 1361 (67%) | 1016 (60%) | 183 (69%) | 2560 (64%) |
| Yes | 674 (33%) | 685 (40%) | 81 (31%) | 1440 (36%) |
| **Matching app** [2] | | | | |
| No | 1989 (98%) | 1556 (91%) | 246 (93%) | 3791 (95%) |
| Yes | 46 (2%) | 145 (9%) | 18 (7%) | 209 (5%) |

[1] Ever been diagnosed. [2] Ever used in the past month.

More than half (56%) the men were sexless during the past month (Table 3). The proportion of sexlessness was 49% for married men and 64% for never-married men (Table 3). Whilst 31% of all participants had only one sex partner, 6% reported having had multiple sex partners in the past month (Table 3). The proportion of married men who did not have sex within marriage but had casual sex in the past month was 6%. The proportion of married men having both casual and non-casual sex was 4%. The proportion of never-married men who had casual sex but did not have non-casual sex was 8% (Table 3). As many as 43% of never-married men and 19% of married men remained sexless in 2019 (Table 3). The percentage of men who had never had sex was 1% among those who were married, 23% among those who had never been married, and 7% among men with other marital statuses (Table 3).

**Table 3.** Sexual behavior in the past month and in 2019 and experience of sex by marital status (*n* = 4000).

| Sexual Behavior | Married (*n* = 2035) | Never Married (*n* = 1701) | Other (*n* = 264) | Total (*n* = 4000) |
|---|---|---|---|---|
| **Number of sex partners in the past month** | | | | |
| 1 | 815 (40%) | 367 (22%) | 72 (27%) | 1254 (31%) |
| 2+ | 108 (5%) | 112 (7%) | 32 (12%) | 252 (6%) |
| 0 (Sexless) | 1000 (49%) | 1088 (64%) | 134 (51%) | 2222 (56%) |
| Missing | 112 (6%) | 134 (8%) | 26 (10%) | 272 (7%) |
| **Sex by type of relationship in the past month** | | | | |
| Non-casual [1] only | 735 (36%) | 287 (17%) | 48 (18%) | 1070 (27%) |
| Both non-casual [1] and casual | 75 (4%) | 62 (4%) | 15 (6%) | 152 (4%) |
| Casual only | 113 (6%) | 130 (8%) | 41 (16%) | 284 (7%) |
| Sexless | 1000 (49%) | 1088 (64%) | 134 (51%) | 2222 (56%) |
| Missing | 112 (6%) | 134 (8%) | 26 (10%) | 272 (7%) |
| **Any sex in 2019** | | | | |
| No | 379 (19%) | 737 (43%) | 87 (33%) | 1203 (30%) |
| Yes | 1495 (73%) | 782 (46%) | 152 (58%) | 2429 (61%) |
| Missing | 161 (8%) | 182 (11%) | 25 (9%) | 368 (9%) |
| **Ever had sex** | | | | |
| Yes | 1977 (97%) | 1216 (71%) | 236 (89%) | 3429 (86%) |
| No | 22 (1%) | 393 (23%) | 18 (7%) | 433 (11%) |
| Missing | 36 (2%) | 92 (5%) | 10 (4%) | 138 (3%) |

[1] Spouse for married men; fiancé or girlfriend/boyfriend for men who have never married and have other marital statuses.

Among married men who had had sex in the past month, 88% had had sex with their spouse, while 7% reported having had sex with a girlfriend/boyfriend, and another 7% reported having had sex with friends with benefits (*sex friend*) (Supplementary Table S2). Among never-married men who had had sex in the past month, 15% reported having had sex with friends with benefits (*sex friend*) (Supplementary Table S3). Paid sex (*fuzoku no hito*) was reported by 11% of never-married men who had had sex in the past month. The corresponding proportion was particularly high (36%) among never-married men without non-casual sex (Supplementary Table S3).

Among married men, older age was positively associated with both sexlessness (RRR 1.08, 95%CI 1.06, 1.09) and casual sex (RRR 1.08, 95%CI 1.05, 1.12) (Table 4). Smokers were less likely than nonsmokers to be sexless. No other variables were associated with sexlessness among married men (Table 4). The variables that were positively associated with casual sex were log income (RRR 1.97, 95%CI 1.12, 3.45), working 60+ h per week (RRR 1.83, 95%CI 1.02, 3.27), and using matching apps (RRR 3.97, 95%CI 1.51, 10.44). Married men who had visited a hospital for fertility counseling and/or treatment were more likely to be sexless (RRR 1.26, 95%CI 0.95, 1.66) and less likely to have casual sex (RRR 0.47, 95%CI 0.22, 1.01) (Table 4).

**Table 4.** Predictors of sexual behavior in the past month for married men in a multivariate multinomial logistic regression model (*n* = 1.642).

| Predictor Variables | Sexless | | Casual Sex | |
|---|---|---|---|---|
| | RRR (95%CI) | *p* | RRR (95%CI) | *p* |
| Age (years) | 1.08 (1.06, 1.09) | <0.001 | 1.08 (1.05, 1.12) | <0.001 |
| Self-rated health (ref: Good) | 1.00 | | 1.00 | |
| Poor | 0.98 (0.69, 1.39) | 0.889 | 0.95 (0.45, 1.99) | 0.894 |
| Depression (ref: No) | 1.00 | | 1.00 | |
| Yes | 1.03 (0.68, 1.57) | 0.885 | 0.51 (0.17, 1.52) | 0.226 |
| Fertility treatment (ref: No) | 1.00 | | 1.00 | |
| Yes | 1.26 (0.95, 1.66) | 0.104 | 0.47 (0.22, 1.01) | 0.054 |
| Smoking (ref: No) | 1.00 | | 1.00 | |
| Yes | 0.75 (0.58, 0.95) | 0.018 | 1.26 (0.78, 2.02) | 0.343 |
| Education (ref: Less than university) | 1.00 | | 1.00 | |
| University or higher | 1.09 (0.86, 1.38) | 0.490 | 0.85 (0.52, 1.40) | 0.528 |
| Job status (ref: Full-time) | 1.00 | | 1.00 | |
| Not full-time | 0.73 (0.49, 1.09) | 0.128 | 1.14 (0.54, 2.41) | 0.725 |
| Log income | 0.95 (0.71, 1.27) | 0.733 | 1.97 (1.12, 3.45) | 0.018 |
| Weekly work hours (ref: <50) | 1.00 | | 1.00 | |
| 50–59 | 1.00 (0.78, 1.30) | 0.977 | 1.07 (0.62, 1.83) | 0.818 |
| 60+ | 0.84 (0.61, 1.17) | 0.309 | 1.83 (1.02, 3.27) | 0.042 |
| Place of residence (ref: Rural) | 1.00 | | 1.00 | |
| Suburban | 0.86 (0.61, 1.20) | 0.370 | 1.76 (0.76, 4.05) | 0.184 |
| Urban | 1.07 (0.78, 1.48) | 0.666 | 1.89 (0.85, 4.22) | 0.118 |
| TikTok use (ref: No) | 1.00 | | 1.00 | |
| Yes | 0.83 (0.53, 1.30) | 0.410 | 0.53 (0.20, 1.45) | 0.218 |
| LINE use (ref: No) | 1.00 | | 1.00 | |
| Yes | 0.93 (0.67, 1.30) | 0.683 | 1.57 (0.70, 3.49) | 0.272 |
| Twitter use (ref: No) | 1.00 | | 1.00 | |
| Yes | 0.95 (0.75, 1.22) | 0.711 | 1.23 (0.74, 2.05) | 0.418 |
| Facebook use (ref: No) | 1.00 | | 1.00 | |
| Yes | 0.90 (0.70, 1.15) | 0.405 | 0.86 (0.51, 1.43) | 0.554 |
| Instagram use (ref: No) | 1.00 | | 1.00 | |
| Yes | 0.85 (0.66, 1.11) | 0.228 | 1.10 (0.65, 1.88) | 0.714 |
| Matching app use (ref: No) | 1.00 | | 1.00 | |
| Yes | 0.75 (0.36, 1.57) | 0.450 | 3.97 (1.51, 10.44) | 0.005 |

CI: confidence interval. RRR: relative risk ratio. *n* = 393 men with any missing values were excluded from the model.

Among never-married men, the following variables were positively associated with sexlessness: age (RRR 1.11, 95%CI 1.08, 1.13), depression (RRR 2.11, 95%CI 1.03, 4.32), and no full-time employment (RRR 1.68, 95%CI 1.10, 2.56) (Table 5). Variables negatively associated with sexlessness were smoking (RRR 0.52, 95%CI 0.35, 0.77), university or higher education (RRR 0.70, 95%CI 0.50, 0.97), log income (RRR 0.59, 95%CI 0.41, 0.85), living in suburban areas (RRR 0.52, 95%CI 0.30, 0.92) or urban areas (RRR 0.53, 95%CI 0.31, 0.90), and Instagram use (RRR 0.58, 95%CI 0.41, 0.83) (Table 5). Having casual sex was positively associated with age (RRR 1.08, 95%CI 1.04, 1.11), not being a full-time employee (RRR 2.14, 95%CI 1.16, 3.95), and the use of matching apps (RRR 3.44, 95%CI 1.81, 6.54); and negatively associated with a university or higher education (RRR 0.59, 95%CI 0.36, 0.97) and Instagram use (RRR 0.55, 95%CI 0.32, 0.95) (Table 5).

The above results in the multivariate multinomial logistic regression models adjusted for all variables were similar to those in the multinomial logistic regression models adjusted only for age (Supplementary Tables S4 and S5). However, the weekly work hours of married men were not significantly associated with casual sex when adjusted only for age (Supplementary Table S4), whereas, after adjustment for other variables, long work hours (60+) were associated with casual sex (Table 4). In addition, for both married and never-married men, having weekly work of 60+ hours was negatively associated with sexlessness (Sup-

plementary Tables S4 and S5), when adjusted solely for age. Nevertheless, the associations were largely attenuated when adjusted for all other variables (Tables 4 and 5).

**Table 5.** Predictors of sex for never-married men in a multivariate multinomial logistic regression model (*n* = 1.263).

| Predictor Variables | Sexless | | Casual Sex | |
|---|---|---|---|---|
| | RRR (95%CI) | *p* | RRR (95%CI) | *p* |
| Age (years) | 1.11 (1.08, 1.13) | <0.001 | 1.08 (1.04, 1.11) | <0.001 |
| Self-rated health (ref: Good) | 1.00 | | 1.00 | |
| Poor | 1.43 (0.87, 2.33) | 0.158 | 1.30 (0.63, 2.67) | 0.477 |
| Depression (ref: No) | 1.00 | | 1.00 | |
| Yes | 2.11 (1.03, 4.32) | 0.041 | 2.00 (0.76, 5.26) | 0.163 |
| Smoking (ref: No) | 1.00 | | 1.00 | |
| Yes | 0.52 (0.35, 0.77) | 0.001 | 0.94 (0.54, 1.64) | 0.837 |
| Education (ref: Less than university) | 1.00 | | 1.00 | |
| University or higher | 0.70 (0.50, 0.97) | 0.030 | 0.59 (0.36, 0.97) | 0.037 |
| Job status (ref: Full-time) | 1.00 | | 1.00 | |
| Not full-time | 1.68 (1.10, 2.56) | 0.017 | 2.14 (1.16, 3.95) | 0.015 |
| Log income | 0.59 (0.41, 0.85) | 0.005 | 1.17 (0.70, 1.97) | 0.549 |
| Weekly work hours (ref: <50) | 1.00 | | 1.00 | |
| 50–59 | 0.98 (0.63, 1.51) | 0.915 | 0.89 (0.45, 1.73) | 0.723 |
| 60+ | 0.70 (0.40, 1.21) | 0.203 | 1.03 (0.46, 2.27) | 0.950 |
| Place of residence (ref: Rural) | 1.00 | | 1.00 | |
| Suburban | 0.52 (0.30, 0.92) | 0.024 | 0.48 (0.21, 1.12) | 0.089 |
| Urban | 0.53 (0.31, 0.90) | 0.019 | 0.78 (0.37, 1.65) | 0.509 |
| TikTok use (ref: No) | 1.00 | | 1.00 | |
| Yes | 0.90 (0.58, 1.40) | 0.638 | 0.76 (0.36, 1.58) | 0.461 |
| LINE use (ref: No) | 1.00 | | 1.00 | |
| Yes | 0.79 (0.48, 1.30) | 0.355 | 1.67 (0.77, 3.62) | 0.194 |
| Twitter use (ref: No) | 1.00 | | 1.00 | |
| Yes | 1.19 (0.83, 1.71) | 0.351 | 0.87 (0.51, 1.48) | 0.601 |
| Facebook use (ref: No) | 1.00 | | 1.00 | |
| Yes | 0.86 (0.61, 1.22) | 0.398 | 0.98 (0.58, 1.66) | 0.944 |
| Instagram use (ref: No) | 1.00 | | 1.00 | |
| Yes | 0.58 (0.41, 0.83) | 0.002 | 0.55 (0.32, 0.95) | 0.031 |
| Matching app use (ref: No) | 1.00 | | 1.00 | |
| Yes | 1.13 (0.69, 1.85) | 0.616 | 3.44 (1.81, 6.54) | <0.001 |

CI: confidence interval. RRR: relative risk ratio. *n* = 438 men with missing values were excluded from the model.

## 4. Discussion

This study is the first to provide quantitative information on both casual sex and sexlessness among men in Japan. The data show that there are sexual relationships outside of intimate and committed relationships, such as with spouses, fiancés, or girlfriends/boyfriends. This could be related to the fact that sexless relationships are common in intimate and committed relationships. A new and interesting finding was that the likelihood of being sexless was lower among never-married men living in suburban and urban areas than among men living in rural areas, even after adjusting for socioeconomic variables. As for the use of matching apps, both married and unmarried men were more likely to have casual sex, suggesting that such tools may facilitate sex in more casual and volatile relationships.

In this study, the proportion of married men who had not had sex in the past year was 19%, and the proportion of missing responses was 8%. In a national survey conducted in Japan in 2007, the corresponding proportion among married men aged 20–59 was 22%, while the proportion of missing responses was 11%. Thus, the proportion of men who have had no sex in the past year was similar in the present and previous studies in Japan [1]. On the other hand, the corresponding proportions seem to be lower in married men in Hong Kong than in Japan, especially for those aged 35 and older. The proportions of men not

having had sex in the past year in Hong Kong were 5.5%, 5.1%, and 17.0% for those in the age ranges of 25–34, 35–44, and 45–59, respectively [21], whereas among the participants in the present study, the proportions were 0%, 5%, 15%, and 28% for those in the age ranges of 20–24, 25–34, 35–44, and 45–54 years, respectively.

In the present study, 49% of the married men remained sexless, whereas 9% had casual sex in the past month. In the 1994 General Social Survey conducted in the United States, the proportion of men and women who had extramarital sex in the past year (not in the past months) was 4.1% and 1.7%, respectively [22]. The authors could not find other comparative data; however, 9% of participants reporting casual (extramarital) sex seemed a high percentage, especially compared with 36% of married men having sex exclusively with their spouse in the past month.

Casual sex was strongly positively associated with the use of matching apps for both married and never-married men. This finding is consistent with a previous study that found a positive association between dating app use and the intention to be unfaithful [23]. Interestingly, a contrast between married and single men was observed in the present study. Among married men, there was no association between SNS use and sexual behavior. Among never-married men, Instagram users were less likely to remain sexless and were less likely to have casual sex. The authors are not aware of any previous studies that have found an association between Instagram use and sexual behavior.

Never-married men living in suburban and urban areas were less likely to be sexless than those living in rural areas. One possible explanation for this difference is the higher population density in urban and suburban areas. Matching apps utilize information on users' geographic locations and help them find sexual partners in nearby areas. However, considering that only 9% of never-married men have used matching apps in the past month, there may be other factors that contribute to the observed regional differences in sexual behavior. Further research using participants' geographic data is warranted to understand geographic differences in sexual behavior in this context. However, there were no such regional differences among married men. In a previous study in Japan, sexless marriages tended to be more prevalent in urban areas (46%) than in rural areas (40%), but the regional difference was not statistically significant [1].

Married men with higher incomes were more likely to have casual sex, whereas no association was found among never-married men. However, sexlessness was not associated with income among married men, while never-married men with higher incomes were less likely to be sexless. Among never-married men, education was negatively associated with both sexlessness and casual sex. Interestingly, these relationships were also observed after adjusting for income and occupational status. This may be because single men with higher educational attainment are more likely to have intimate and committed relationships. In the 2015 National Fertility Survey, among single men aged <35 years, the proportion having an intimate and committed relationship was 18% in those with less than university education and 23% in those with university or higher degree (calculated by the authors from the report [24] p. 219). The corresponding proportions in the current participants (aged 20–54 years) were 19% and 26%, respectively (results not shown). Men with higher and lower levels of education may have contrasting sexual norms [25]; this warrants further investigation. Japanese men with lower educational attainment are more likely to remain unmarried [26]. Therefore, how educational differences in sexual behavior are related to differing marriage rates by educational status requires further exploration.

Long working hours (60+ hours per week) in married men was positively associated with casual sex, while long working hours was not significantly associated with being sexless. This is in contrast to the finding by Genda and Kawakami [7] that long working hours of salaried workers was associated with lower coital frequency. One potential explanation for the discrepancy between the present study and Genda and Kawakami [7] may be that, in the present study, participants were categorized into three groups—sexless, non-casual sex, or casual sex—while Genda and Kawakami used the frequency of sex, regardless of the type of sex partner. A potential but highly speculative interpretation of

the present finding is that men who are less satisfied with their marriages tend to work long hours away from home and have casual sex. However, the authors are aware that many men work long hours without having casual sex.

Eight percent of participants reported having suffered from depression, which is similar to the lifetime prevalence of major depressive disorder (6.1%) in the World Mental Health Japan Survey 2002–2006 [27]. Never-married men with depression were more than twice as likely to be sexless than those without depression. This may be explained in part by the side effects of antidepressants, as they reduce libido [28]. However, considering that depression in married men was not associated with sexlessness, it is speculated that depression in never-married men is associated with a lower chance of having sex due to less active communication with people. The links between mental health, communication, and sexual behavior may also be an important focus of future research.

The use of online recruitment and surveys is an advantage of this study. Web-based questionnaires allow participants to respond using their smartphones, tablets, PCs, or other devices without interacting with an interviewer. The authors believe this made it easier for participants to report on their sexual activities. In addition, recruiting participants from the registered monitors of a survey company was quick and could be performed at a much lower cost than the traditional sampling method. A limitation of the present study is that it relies on self-reporting, which may not be accurate. However, self-reporting is one of the most reliable ways of obtaining information about people's sexual activities. Another limitation is that the participants were not recruited using a nationwide random sample. The sexual behavior of our participants may differ from that of the general population in Japan if the characteristics of the participants differ from those of the national population. It is also possible that their sexual behavior in the past month was influenced by COVID-19. Further research is needed to investigate whether the current results can be replicated in other study populations in Japan and other countries.

## 5. Conclusions

The aims of this study were to (1) collect quantitative data on sexual behavior by relationship type and (2) explore the correlates of casual sex and sexlessness among men in Japan. Using a web-based questionnaire, the authors obtained information on the sexual activity of 4000 men aged between 20 and 54 years residing in Japan. In married men, higher income and long working hours were positively associated with having casual sex. Never-married men with a lower educational status and without full-time jobs were more likely to report casual sex. Never-married men in rural areas were more likely to be sexless than those in urban and suburban areas. Matching app use was strongly associated with casual sex among both married and never-married men, suggesting that such tools may facilitate sex outside of committed and intimate relationships. Sexual behavior may be one of many pathways through which the socioeconomic status and social environment impact fertility and reproduction. The relationships between social and economic factors and low fertility in this setting can be better understood if we gain more insight into sexual behavior.

**Supplementary Materials:** The following are available online at https://www.mdpi.com/article/10.3390/sexes3020020/s1: Supplementary Material 1: Selected questions from the cross-sectional questionnaires of the IITEF project. Responses of the listed questions were used in the analyses. Table S1: Number of participants by 5-year age category and eight geographic regions. These numbers are relative to the corresponding proportions of the male population in Japan in the 2015 National Census; Table S2: Reported types of sex partners of married men in the past month by non-casual sex (*n* = 923); Table S3: Reported types of sex partners of never-married men by non-casual sex (*n* = 479); Table S4: Predictors of sexual behavior in the past month for married men. Each variable was analyzed in separate multinomial logistic regression models while adjusting for age; Table S5: Predictors of sexual behavior in the past month for never-married men. Each variable was analyzed in separate multinomial logistic regression models while adjusting for age.

**Author Contributions:** Conceptualization, S.K., Y.M. and M.A.; Methodology, S.K., Y.M. and M.A.; Formal Analysis, S.K. and F.K.; Investigation, S.K., Y.M. and M.A.; Data Curation, S.K. and F.K; Writing—Original Draft Preparation, S.K.; Writing—Review and Editing, S.K., Y.M., M.A. and F.K.; Project Administration, S.K. and F.K.; Funding Acquisition, S.K. All authors have read and agreed to the published version of the manuscript.

**Funding:** This research was funded by the Japan Society for the Promotion of Science (JSPS) Topic-Setting Program to Advance Cutting-Edge Humanities and Social Sciences Research, Global Initiatives, Grant Number JSPS00119217822 (P.I: S.K.).

**Institutional Review Board Statement:** The study was conducted in accordance with the guidelines of the Declaration of Helsinki and approved by the Institutional Research Ethics Committee of the Graduate School of Medicine and Faculty of Medicine, University of Tokyo (2020022NI, 20 May 2020).

**Informed Consent Statement:** Informed consent was obtained from all subjects involved in the study, after they were made aware of the objectives of the study, the items in the questionnaires, and the honorarium they would receive upon completion of the survey. Subjects were also informed that they could withdraw from the study during or after completion of the questionnaire; however, their responses would not be excluded once the questionnaire was completed.

**Data Availability Statement:** The data presented in this study are available upon request from the corresponding author. The data are not publicly available due to ethical issues.

**Acknowledgments:** The authors acknowledge Keisuke Hattori and Kyoko Takeuchi for their help with creating the questionnaire. We also thank all the participants of this study.

**Conflicts of Interest:** The authors have no conflicts of interest to declare.

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
