# Peer review of "Casual Sex and Sexlessness in Japan: A Cross-Sectional Study"

_sexes, doi:10.3390/sexes3020020_

Round 1

Author Response

Response to reviewers’ comments

We appreciate the reviewers for their insightful comments, which have helped us improve the manuscript greatly. We have provided our responses to each of their comments and described the revisions made to the manuscript below. The page and line numbers correspond to the clean version of the revised manuscript (without tracked changes).

REVIEWER 1

  • A brief summary

This study explores the correlates of casual sex and sexlessness relationships.

The results show that sexlessness is high among Japanese married men (56%), while 11% had casual relationships and 31% had non-casual sex. Some trends were identified, with casual sex being higher among married men with higher income and long working hours (60+ hours per week) and among never-married men with a lower educational status and without full-time jobs. Sexlessness was higher among never-married men in rural areas than those in urban and suburban areas.

The authors mentioned that to date no quantitative study in Japan has examined sexual activities by different types of relationships, namely casual or non-casual. Nevertheless, the relevance of this data or the potential social impact of such knowledge is not argued.

Response:

- Thank you for your comments. We have added several sentences in Introduction and Conclusions to describe the relevance of the data and the potential social impact as follows:

Identifying the correlates of sexual behavior can help us understand the social structure that contributes to widespread sexlessness among married couples in this setting. Considering that pregnancy (but not childbirth) often precedes marriage in Japan [16], sexual behavior of single men is expected to impact the rates of marriage and fertility in the country. (Page 3, Lines 106–110)

Sexual behavior may be one of many pathways through which socioeconomic status and social environment impact fertility and reproduction. The relationships between social and economic factors and low fertility in this setting can be better understood if we gain more insight into sexual behavior. (Page 12, Lines 411–414)

  • General concept comments
  1. Abstract and keywords:
  • Smoking I presented as a keyword, when it is not presented in the lit review, or argued how this is relevant for the study

Response:

- Thank you for pointing this out. We have deleted the term from the list of keywords.

  • Authors mention “Sexual behavior may mediate the links between social environment,

socioeconomic status, and reproduction.” Nevertheless, the articles does not focus on preproduction, and as data are presented it looks like the social, economic and health issues presented mediate sexual behavior. In this same line, the abstract does not mention health aspects, which are presented in the results and discussion.

Response:

- Thank you for pointing out this important issue. We have revised the Abstract as follows:

Regarding never-married men: those of lower educational status and without full-time jobs were more likely to report casual sex; those in rural areas were more likely to be sexless than those in urban and suburban areas; and those with depression were more likely to be sexless that those without depression. Matching app use was strongly associated with casual sex among married and never-married men, suggesting that such tools may facilitate sexual activity outside committed and intimate relationships. Sexual behavior is closely linked to one’s social and economic environment and health status. (Page 1, Lines 19–25)

  1. The literature review is shallow and not very well linked to the topic of the article “Correlates of Casual Sex and Sexless Relationships in Japan:”
  • It starts by presenting sexless marriage and low fertility, and explaining the link between coital interactions and low fertility. Nevertheless, the relevance of the latter (fertility and low coital interaction) to the objective of the article (correlates of casual sex –sexless relationships) is not explained.

Response:

- Thank you for the suggestion. We have added several sentences to the Introduction that explain the relevance of coital interactions and low fertility. The revised text is as follows:

Given that sexless marriage and low fertility may be interrelated, examining what factors contribute to sexlessness and casual sex is important. Sexless relationships may already be present among unmarried couples. We speculate that sexlessness within intimate and committed relationships may be partly compensated by casual sex. Thus, we explored which social, economic, and health-related factors are associated with sexlessness and casual sex among married and never-married men.  (Page 2, Lines 81–86)

  • P.2 lin 57→A survey is mentioned in which 3.2% of male respondents and 2.1% of female respondents had one or more casual sex partners. At the end of that paragraph this survey is used to justify casual sex as “relatively common”, when the percentages do not show that. Percentages in the other survey mentioned do show this commonality, but not the first.

Response:

- Thank you for pointing this out. Accordingly, we have corrected the sentence as follows:

The latter survey, but not the former, suggests that casual sex may be relatively common, which contrasts with the prevalence of sexless marriages in Japan. (Page 2, Lines 62–64)

  • P.2 line 77→the following statement is made “”As casual sex is more com-mon among men than women [9]”. Nevertheless, the sentence does not make reference to Japan, and the cited reference is a Survey, which is not a scientific paper. Thus, the statement should be better precised.

Response:

- As per your suggestion, we have added two references and modified the relevant sentence as follows:

As casual sex is more common among men than women in Japan [9] and in other countries [10,11], this study targeted only men aged 20–54 years. (Page 2, Lines 78–80)

  • From line 69 to line 78 in the lit review the authors present questions of their own survey as well as characteristics of the sample, which normally belong in the methodology section.

Response:

- In the Introduction section, we have presented the research question, and not actual questions from the questionnaire. We agree with your point that the description of the seven different types of relationships may belong in the Methods section. However, we judged that this description is required in the Introduction section to explain the novelty of our study.

  • Line 82 “Therefore, this study analyzed the relationship between SNSs and matching apps and casual sex.” Why is this relevant to the objective of the article?

- Thank you for your question. To clarify the relevance of examining correlates of sexual behavior, we have added several sentences in the Introduction as follows:

Given that sexless marriage and low fertility may be interrelated, examining what factors contribute to sexlessness and casual sex is important. Sexless relationships may already be present among unmarried couples. We speculate that sexlessness within intimate and committed relationships may be partly compensated by casual sex. Thus, we explored which social, economic, and health-related factors are associated with sexlessness and casual sex among married and never-married men. We assumed that sex within intimate and committed relationships is more likely to result in pregnancy and marriage, which would lead to reproduction. By contrast, we assumed that casual sex is less likely to result in reproduction. By examining factors associated with casual sex and sexlessness, we aimed to understand who among the Japanese people are more likely to engage in sexual activity that is less likely to result in reproduction. (Page 2, Line 81–91)

  • Line 83 “Socioeconomic status and health-related factors were included in the analyses, as they were also assumed to be associated with sexual behavior.” No quotes or scientific background have been presented to argue why these factors are relevant to the objective of the study.

Response:

- Thank you for pointing this out. To clarify our meaning, we have added several sentences to the last paragraph of the Introduction section:

Given that sexless marriage and low fertility may be interrelated, examining what factors contribute to sexlessness and casual sex is important. Sexless relationships may already be present among unmarried couples. We speculate that sexlessness within intimate and committed relationships may be partly compensated by casual sex. Thus, we explored which social, economic, and health-related factors are associated with sexlessness and casual sex among married and never-married men. We assumed that sex within intimate and committed relationships is more likely to result in pregnancy and marriage, which would lead to reproduction. By contrast, we assumed that casual sex is less likely to result in reproduction. By examining factors associated with casual sex and sexlessness, we aimed to understand who among the Japanese people are more likely to engage in sexual activity that is less likely to result in reproduction. (Page 2, Line 81–91)

  1. The methodology is clearly presented. Nevertheless, some issues need to be clarified:
  • in the statistical analysis (line 178) the authors state that “If he [a male respondent] had both casual and non-casual sex, he was categorized as having non-casual sex in the statistical analyses “. Authors do not argue why this decision was made, instead of creating a separate category for men in both types of relationships.

Response:

- Thank you for the suggestion. We have added a sentence to explain our reasoning as below:

We did not set a category which included having both casual and non-casual sex because of the small sample size. (Page 5, Lines 203–204)

  • Line 193 “Predictor variables for never-married men were the same as those for married men, except that fertility treatment was not included”. It is the first time that the authors “fertility treatment” in the paper”. Why is it relevant to the objective of the study, which does not have to do with fertility?

Response:

- Thank you for pointing this out. In one of our previous studies targeting Japanese women, we found that those who had received fertility treatment were having less sex compared to those who had never received fertility treatment. Therefore, we included the variable in the analysis of the present study. Accordingly, we have presented fertility treatment in the Introduction section, with the relevant citation, as follows:

Furthermore, fertility treatment among married couples in Japan was associated with less frequent sex [15]. (Page 3, Lines 101–103)

  • Line 195 “These predictor variables were selected because health status, health-related behaviors, and socioeconomic status are expected to influence motivation, libido, and/or the ability to have sex.” The authors make this statement without having covered these in the lit review.

Response:

- Thank you for pointing this out. We have added the relevant information to the last paragraph of the Introduction section as follows:

Given that sexless marriage and low fertility may be interrelated, examining what factors contribute to sexlessness and casual sex is important. Sexless relationships may already be present among unmarried couples. We speculate that sexlessness within intimate and committed relationships may be partly compensated by casual sex. Thus, we explored which social, economic, and health-related factors are associated with sexlessness and casual sex among married and never-married men. We assumed that sex within intimate and committed relationships is more likely to result in pregnancy and marriage, which would lead to reproduction. By contrast, we assumed that casual sex is less likely to result in reproduction. By examining factors associated with casual sex and sexlessness, we aimed to understand who among the Japanese people are more likely to engage in sexual activity that is less likely to result in reproduction.

Casual sex may be facilitated by social networking services (SNSs) and matching apps. In 2020, more than 85% of individuals aged 20–59 years in Japan owned smartphones [12]. Because SNSs and matching apps can be installed and used on smartphones, it makes it easier to meet casual sex partners. Therefore, this study analyzed the relationship between SNSs and matching apps and casual sex. Socioeconomic status and health-related factors were included in the analyses, as they were also considered to be associated with sexual behavior. In a previous study using the JGSS data, long working hours were associated with less frequent sex [7]. Additionally, health-related factors have been shown to be associated with sexual behavior and functioning–depression and erectile dysfunction often present together in men [13,14]. Furthermore, fertility treatment among married couples in Japan was associated with less frequent sex [15]. Consequently, depression and fertility treatment were included in the analyses. This study aimed to: (1) collect quantitative data on sexual behavior by relationship type; and (2) explore the correlates of casual sex and sexlessness among men in Japan. Identifying the correlates of sexual behavior can help us understand the social structure that contributes to widespread sexlessness among married couples in this setting. Considering that pregnancy (but not childbirth) often precedes marriage in Japan [16], sexual behavior of single men is expected to impact the rates of marriage and fertility in the country. (Page 2, Line 81 to Page 3, Line 110)

  • Participants are asked about masturbation, but this has not been presented before, or its influence on casual and sexless relationships.

Response:

- Thank you for pointing this out. We have deleted the data and description regarding masturbation from the main text and Table 3.

  1. Ethical aspects. Authors make reference to participant having signed a consent form as part of the survey. Nevertheless, they do not make reference to which information were the participants given (objective of the study, type of participation, possibilities for withdrawing...). Nor they give information of an ethics committee approving the study.

Response:

  • The information that the participants were provided before signing informed consent are described in the Informed Consent Statement section on Page 13 as follows:

Informed Consent Statement: Informed consent was obtained from all subjects involved in the study, after they were made aware of the objectives of the study, the items in the questionnaires, and the honorarium they would receive upon completion of the survey. Subjects were also informed that they could withdraw from the study during or after completion of the questionnaire; however, their responses would not be excluded once the questionnaire was completed.

  • Information regarding the Ethics Committee approval of this study is described in the Institutional Review Board Statement section on Page 13.
  •  

Institutional Review Board Statement: The study was conducted in accordance with the guidelines of the Declaration of Helsinki and approved by the Institutional Research Ethics Committee of the Graduate School of Medicine and Faculty of Medicine, University of Tokyo (2020022NI, May 20, 2020).

  1. Results.
  • The issue of fertility is mentioned again. The survey they present is part of a wider study which considers fertility, but fertility does not seem to be relevant to the study of Correlates of Casual Sex and Sexless Relationships in Japan, and its relevance has not been argued in the paper.

Response:

- Thank you for this comment. We have now added a sentence to the Introduction section, explaining the potential importance of fertility treatment as follows:

Furthermore, fertility treatment among married couples in Japan was associated with less frequent sex
[15]. (Page 3, Lines 101–103)

  • Line 221: “The proportion of married men having both casual and non-casual sex was 4%.” This information is presented while in the methodology authors mentioned that “If he [a male respondent] had both casual and non-casual sex, he was categorized as having non- casual sex in the statistical analyses “ Something similar happens in the discussion (line 295- 300)

Response:

- We have deleted the potentially misleading sentence (“Overall, 9% … having casual sex.”).

  • Results on Masturbation are presented, but how these are relevant to the topic of study has not been presented before.

Response:

  • We have deleted the data and description regarding masturbation from the main text and Table 3.

  1. Discussion
  • Line 276 “The data show that there are many sexual relationships outside of intimate and

committed relationships, such as with spouses, fiancés, or girlfriends/boyfriends.” Nevertheless, in the results the authors comment that “Overall, 9%(n = 75 + 113 = 188) of married men reported having casual sex. The proportion of never-222 married men who had casual sex but did not have non-casual sex was 8% (Table 3). These two statements sound contradictory, or the word “many” is not appropriate.

Response:

  • Thank you for the suggestion. We have deleted the word “many.” Now, it reads:

The data show that there are sexual relationships outside of intimate and committed relationships, such as with spouses, fiancés, or girlfriends/boyfriends. (Page 10, Lines 301–303)

  •  
  • The authors continue by saying “this could be related to the fact that sexless marriages are common.” Nevertheless, according to their own results, it was more common for never-married men “The proportion of sexlessness was 49% for married men, whereas it was 64% for never- married men (Table 3).”

Response:

  • We thank the reviewer for the insightful comments. We have modified the sentence as follows:

This could be related to the fact that sexless relationships are common in intimate and committed relationships. (Page 10, Lines 303–304)

  •  
  • The discussion is based on many factor that have not been presented & contextualized in the lit review (e.g. depression, long working hours...)

Response:

  • Thank you for the suggestion. We have added the relevant information regarding depression, long working hours, and fertility treatment, to the Introduction section.

  1. References. Many of the references are not peer reviewed papers indexed in scientific databases.

Response:

- Thank you for the suggestion. We have added eight new references that are peer reviewed papers indexed in scientific databases.

REVIEWER 2

The present work deals with a sensitive topic and not always easy to be studied.

The study reveals methodological rigor, contemplating a considerable sample of Japanese men (4000).

In general, it is well structured and written and in accordance with the rules of the journal. Notwithstanding the formal quality of the article, my main question concerns the usefulness of the study for practice.

In this sense, I believe that the authors should better explore the pertinence and relevance of the study. Therefore, I consider that the introduction needs greater depth and theoretical robustness, objectively identifying the practical implications associated with the object of study. In line with this idea, I believe that the discussion and conclusions section should also better explain the practical implications of the study, as well as clues for future research.

Response:

  • Thank you for pointing out this important issue. Keeping your comments in mind, we have added a few sentences to the Introduction section to elaborate on the practical implications of our study, as follows:

Identifying the correlates of sexual behavior can help us understand the social structure that contributes to widespread sexlessness among married couples in this setting. Considering that pregnancy (but not childbirth) often precedes marriage in Japan [16], sexual behavior of single men is expected to impact the rates of marriage and fertility in the country. (Page 3, Lines 106–110)

  • We have also added a sentence to the Conclusions section as follows:

The relationships between social and economic factors and low fertility in this setting can be better understood if we gain more insight into sexual behavior. (Page 12, Lines 414–414)

Reviewer 2 Report

The present work deals with a sensitive topic and not always easy to be studied.

The study reveals methodological rigor, contemplating a considerable sample of Japanese men (4000).

In general, it is well structured and written and in accordance with the rules of the journal. Notwithstanding the formal quality of the article, my main question concerns the usefulness of the study for practice.

In this sense, I believe that the authors should better explore the pertinence and relevance of the study. Therefore, I consider that the introduction needs greater depth and theoretical robustness, objectively identifying the practical implications associated with the object of study. In line with this idea, I believe that the discussion and conclusions section should also better explain the practical implications of the study, as well as clues for future research.

Author Response

(The authors gave the same response as above.)

Round 2

Reviewer 2 Report

The manuscript is now much more complete and the authors have integrated all the suggestions.